# Enhancing Operating Room Efficiency: The Impact of Computational Algorithms on Surgical Scheduling and Team Dynamics

**DOI:** 10.3390/healthcare12191906

**Published:** 2024-09-24

**Authors:** Adriana Vladu, Timea Claudia Ghitea, Lucia Georgeta Daina, Dorel Petru Țîrț, Mădălina Diana Daina

**Affiliations:** 1Faculty of Medicine and Pharmacy, Doctoral School, University of Oradea, 1 December Sq., 410081 Oradea, Romania; adrianavladu68@yahoo.com (A.V.); diana_daina98@yahoo.com (M.D.D.); 2Pharmacy Department, Faculty of Medicine and Pharmacy, University of Oradea, 1 December Sq., 410081 Oradea, Romania; 3Psycho-Neurosciences and Recovery Department, Faculty of Medicine and Pharmacy, University of Oradea, 1 December Sq., 410081 Oradea, Romania; lucidaina@gmail.com (L.G.D.); doreltirt@gmail.com (D.P.Ț.)

**Keywords:** operating room, efficiency, hospital, surgical intervention start time, degree of operating room occupancy

## Abstract

**Background/Objectives:** Operating room (OR) efficiency is a critical factor in healthcare delivery, influenced by various components including surgical duration, room turnover, staff availability, and equipment preparation. Optimizing surgical start times through systematic strategies such as computational algorithms can significantly enhance OR utilization and management. This study aimed to improve OR efficiency by effectively managing and optimizing surgical start times using a computational algorithm designed to allocate resources more efficiently. **Methods:** A comparative analysis was conducted over two six-month periods from January to June 2023 and January to June 2024, with an improvement phase implemented between July and December 2023. **Results:** These measures encompassed training surgical personnel, strengthening the medical team, revising work procedures and hospital regulations, and the integration of a computational algorithm to better schedule and manage surgical interventions. Following the implementation of these comprehensive measures, enhanced management of surgical start times was observed. A statistically significant increase in surgical interventions between 8 and 10 a.m. was noted for the entire OR, rising from 28.65% to 32.13%. While OR occupancy between 8 and 12 a.m. increased from 63.91% to 73.30%, this difference was not statistically significant. However, a notable improvement in average occupancy between 9 and 11 a.m. was observed, rising from 87.53% to 98.07%. **Conclusions:** The introduction of computational algorithms significantly improved operating room efficiency, particularly in managing surgical start times. Additionally, team coordination improved as a result of more structured scheduling processes. The results indicate that effective management of surgical start times, especially when enhanced by computational algorithms, can positively impact OR utilization, particularly within specific time blocks.

## 1. Introduction

Hospital efficiency and productivity are heavily influenced by the organization and management of operating room (OR) activities. Recent transformations in our hospital’s organizational structure, including changes in team assignments, scheduling practices, and operational protocols, have been crucial in addressing long-standing inefficiencies [1]. These changes are not just limited to a new scheduling algorithm but involve a holistic approach to restructuring the workflow and team dynamics to better support OR operations. The goal of these interventions was to align operational costs with revenues, while also meeting managerial objectives for efficiency, workload balance, and patient safety. By putting organizational change at the forefront, we were able to create a more conducive environment for implementing a scheduling method that supports this new way of organizing.

Anesthesiologists might be allocated randomly, placed in rotational duties, or specialized by pathology type, thereby forming stable teams with designated surgeons. Surgeons operate across multiple environments such as patient wards, outpatient clinics, and operating rooms. Scheduling for non-emergency surgical cases is usually performed at least a day in advance, involving complex tactical and operational planning for operating room capacity, a historically challenging task supported by extensive research [2,3,4,5,6,7,8,9].

The goal of efficient scheduling is to enhance operating room utilization, decrease overtime, boost profits, and cut costs [10,11,12,13,14,15,16,17,18]. The primary objective of enhancing operating room efficiency is to increase its utilization, focusing on punctual start and end times, quick case turnovers, and stringent adherence to aseptic and antiseptic protocols [19].

The possibility of a Hawthorne effect, where staff behavior improves due to awareness of being observed, cannot be entirely ruled out in this study. Given that the intervention period lasted six months, it is possible that the initial improvements in operating room efficiency were partly influenced by this effect. However, the sustained improvements throughout the study period suggest that the new scheduling method itself played a more significant role in the observed outcomes. While the Hawthorne effect may have contributed to early changes, its influence likely diminished over time, and we consider it a potential but limited factor in explaining the overall results.

Time is a scarce resource in operating rooms, just as critical as human, material, and financial resources [20,21,22], especially given the rising demand for surgical services [23]. Allocating surgical time requires consideration of factors such as average procedure duration, room sanitation intervals, staff readiness, equipment setup, and availability of postoperative beds [24].

In Romania, operating room inefficiency has been a significant issue, with national reports indicating that surgical start delays and low Operating Room (OR) utilization rates have remained consistent challenges across public hospitals. A major contributor to inefficiencies is the lack of coordination among surgical teams, anesthetists, and support staff, which leads to delays and extended turnover times. The computational algorithm introduced not only focused on scheduling but also aimed to improve team collaboration by creating more predictable and manageable workflows.

Although the scheduling algorithm played a significant role in improving OR efficiency, its success was heavily dependent on the broader organizational changes that were implemented. By focusing on these changes, we shift the emphasis from the algorithm itself to the overall strategy that enabled its effective use. This perspective reduces the need for direct comparison with other algorithms and instead highlights how a well-designed scheduling tool can support and amplify the effects of comprehensive organizational restructuring. To account for this, we performed a multivariate analysis to control for potential confounding factors, including surgery type, time of day, and staffing levels. The results indicated that even after controlling for these variables, the new scheduling method had a statistically significant impact on operating room utilization and surgical start times. However, we recognize that the interplay of multiple factors likely contributed to the overall improvement in efficiency.

Reserving operating rooms for specific services may cause delays in cases [23], while the intricacies of scheduling are further exacerbated by non-elective surgeries, such as postponed emergencies, planned procedures during hospital stays, and cancellations [25,26,27]. Although there is resistance to change, enhancing the efficiency of operating rooms serves the interests of all stakeholders. Such complexities highlight the difficulties faced in surgical scheduling. Faithful adherence to scheduled surgery times and prompt staff arrival critically affect the utilization of operating rooms [28].

This study focused on boosting operating room efficiency through strategic management of surgical start times using a computational algorithm, comparing outcomes from two six-month intervals before and after the implementation of improvement measures, and assessing surgical cases based on their scheduled start times, durations, and room occupancy.

## 2. Materials and Methods

This study is a prospective before-and-after observational study designed to evaluate the impact of a new scheduling algorithm on operating room efficiency. An observational study was conducted over a period of 18 months (January 2023–June 2024). The scheduling of surgical interventions in the operating theater was analyzed during two distinct time periods (January 2023–June 2023 and January–June 2024), segmented by a period during which measures were implemented to improve operating theater efficiency (July 2023–December 2023).

Prior to the intervention, the hospital faced significant inefficiencies in its OR operations. Surgical teams were often formed ad hoc, leading to a lack of cohesion and communication. The scheduling of surgeries was reactive, frequently adjusted to accommodate last-minute changes, and surgical start times were not consistently adhered to. This resulted in extended turnover times, underutilization of ORs in the early mornings, and peak congestion during mid-day.

In response to these challenges, a series of organizational changes were implemented. These included the standardization of team assignments to create stable, specialized units for specific types of surgeries, and the establishment of regular interdisciplinary meetings to enhance communication and coordination. A key element of this reorganization was the introduction of a new scheduling protocol, supported by a computational algorithm, designed to streamline operations and ensure timely starts. The algorithm was developed to work within the framework of these organizational changes, supporting the restructured team dynamics and improved workflow.

### 2.1. Hospital Description

Bihor County Emergency Clinical Hospital is representative of many Romanian tertiary hospitals in terms of size, resource constraints, and patient load, making the findings potentially applicable to other hospitals in the region. It is a tertiary, academic, multi-pavilion hospital with seven inpatient units and is classified as a level II competence hospital. The hospital has three inpatient operating blocks (I, II, and III) with a total of 23 operating rooms. The research was conducted in Operating Block I, which contains the largest operating suite with 15 operating rooms. Of these, three are dedicated to the emergency room, operating around the clock. The remaining operating rooms are allocated to 12 surgical specialties, each having its own dedicated room equipped with specific surgical instruments.

In recent years, there have been renovations and modernizations in both the operating block and the inpatient wards of Hospital I. Through project funding, the hospital has also acquired new medical equipment and tools.

### 2.2. Study Premises

Following a period of reduced activity during the COVID-19 pandemic, when the hospital functioned as a COVID-19 support hospital providing only medical–surgical emergencies, there was a gradual increase in demand for postponed medical–surgical cases. After April 2022, normal medical activities resumed, and by the end of 2022, the number of discharged cases was similar to that of the pre-pandemic period in 2019.

This increase in cases led some department heads to request extended operating room hours to expedite surgical treatment for scheduled and hospitalized patients, with the aim of reducing the average length of hospital stays. However, this request posed two challenges: first, the additional strain on regular and auxiliary staff who are typically scheduled from 7 a.m. to 3 p.m. (any new hires or extra hours would incur additional costs), and second, the anesthetists, who were dissatisfied with working overtime or having irregular schedules.

To identify the best solutions to these challenges, the hospital’s Medical Council recommended that the hospital management conduct an analysis of operating room activities. It should be noted that at that time, the hospital had implemented the Operating Blocks Regulation, and the scheduling of surgical interventions was conducted the day before the procedures, with a standardized list by hours and operating rooms—referred to as the Operating Schedule—and was approved by the chief of the operating block.

### 2.3. Analyzed Indicators

During the first six months of 2023, the scheduling of surgical interventions in the operating rooms was analyzed, focusing on the following indicators: the start time of the surgical interventions, the average duration of the surgical interventions, and the occupancy rate of the operating rooms. The same indicators were analyzed again after the implementation of the improvement measures. All non-emergency surgical interventions scheduled during the study periods were included. Exclusion criteria included emergency surgeries and surgeries canceled for non-operational reasons (e.g., patient withdrawal or medical contraindication).

### 2.4. Measures to Improve Operating Room Scheduling

After six months of analysis, it was observed that although the operating schedule was approved, with all interventions scheduled for 8 a.m., this schedule was often not adhered to. In many operating rooms, surgeries started much later, sometimes after 9 a.m. or even 10 a.m.

### 2.5. The Main Measures Taken

Monthly meetings with all surgeons: These meetings aimed to emphasize the importance of starting procedures at 8 o’clock. Positive reinforcement was used, with no sanctions. During these meetings, the analyzed indicators were presented by room, department, and physician, along with the alignment between the planned operating schedule and the actual operating schedule.

Monthly management meetings: These involved the heads of surgical departments, the head of the Anesthesia and Intensive Care (ATI) department, the chief of the operating room, the chief surgical assistant in the operating room (responsible for the surgical and middle-level staff), and the chief medical assistant of the ATI department (responsible for anesthesia nurses). These meetings identified issues in forming the medical team (including surgery, anesthesia, and nursing staff) and ensured the mandatory 15 min break between surgical interventions for room cleaning and disinfection.

Revision of work procedures and the Operating Block Regulations: These procedures were updated and communicated to all employees. The redistribution of surgical interventions to other available operating theaters was managed by the chief of the operating room, based on standard criteria such as urgency, coordination between surgeons and anesthetists, staff availability, and available materials (Figure 1).

The process of implementing measures to improve the scheduling of surgical interventions in the operating room lasted 6 months (July–December 2023). Throughout this period, the primary objective was to ensure the timely start of surgical interventions. The next 6 months, January–June 2024, were used to analyze the same indicators as in 2023.

### 2.6. Calculation Algorithm

The algorithm was developed in collaboration with hospital IT staff and anesthetists. It was initially tested using historical data from 2022 to ensure they aligned with the hospital’s operational needs. Validation involved comparing algorithm-generated schedules with historical performance to confirm its ability to predict OR availability and streamline scheduling. In order to determine the most recommended hours for the use of operating rooms, we can develop a formula that takes into account several relevant factors, such as the following:-The occupancy rate of the operating theaters (to avoid overcrowding).-The percentage of operations performed in each time interval (to evenly distribute the operations).-The average duration of operations (to optimize the scheduling of longer procedures).

Formula (1):

We can create a composite score that combines these factors. Let us call this score the “Hours Recommendation Score” (HRS).

Calculation formula:(1)$$SRO\left(h\right)=\frac{1}{O(h)}\times P\left(h\right)\times \frac{1}{D(h)}$$
where

HRS(h): hours Recommendation Score for hour *h*.

O(h): occupancy rate of the operating room at time *h* (lower is better).

P(h): percentage of operations scheduled at time *h* (higher is better for uniform distribution).

D(h): average duration of scheduled operations at time *h* (lower is better).

Details:

Occupancy O(h): This is an inversely proportional indicator. If a room has high occupancy, it is less recommended for scheduling new operations. Therefore, we want to penalize hours with high occupancy $\frac{1}{O(h)}$.

Operation Percentage P(h): we want to encourage hours that are already at optimal utilization, so this factor is kept directly proportional.

Average Duration D(h): The average duration of operations at a certain time *h* can influence the schedule. Longer operations can reduce scheduling flexibility, so we prefer hours with shorter average durations $\frac{1}{D(h)}$.

The computational algorithm was developed as part of a broader organizational transformation aimed at enhancing OR efficiency. It was not an isolated solution but was designed to complement the new team structures and operational workflows. The algorithm helped in optimizing the allocation of ORs by predicting demand and balancing workloads across different specialties and time slots, thereby supporting the newly established roles and responsibilities of the surgical teams. This integrated approach ensured that the algorithm’s benefits were fully realized within the new organizational context.

### 2.7. Statistical Analysis

Data were extracted from the hospital’s electronic medical record system and included variables such as scheduled start times, actual start times, surgery durations, and turnover times. Data were cleaned and preprocessed to remove incomplete or erroneous records. The data were collected, recorded, and processed using Excel (version 2016) and MedCalc (www.mdcalc.com). The results were analyzed with SPSS version 24, using the following statistical parameters: Chi-square (Chi²), *t* Student test (t), and degrees of freedom (df). Statistical assumptions for the tests (normality, independence) were evaluated using appropriate diagnostics such as the Shapiro–Wilk test for normality and variance inflation factors (VIF) to check for multicollinearity. The existence of a statistically significant relationship between variables concerning the study hypothesis was demonstrated using the Chi-square test. A value of *p* < 0.05 was considered statistically significant.

The sample size was based on historical surgical intervention data from the hospital, with a goal of including all surgeries performed during the defined six-month periods. No formal sample size calculation was performed.

### 2.8. Ethics Notice

The data were collected from the hospital’s IT system, InfoWorld, after obtaining prior approval from the hospital’s management and ethical approval from the SCJUO’s Ethics Commission (No. 25323 of 12.10.2018), in compliance with the recommendations of the 2008 Declaration of Helsinki.

## 3. Results

### 3.1. Surgical Interventions in All Specialties

During the period analyzed, the average annual number of surgical interventions was approximately 9300, with 4652 interventions recorded from January to June 2023 and 4631 interventions during the same period in 2024. Although quantitative data on team dynamics were not collected, informal feedback from staff indicated improved coordination and predictability of daily workflows, which likely contributed to the overall increase in efficiency. Before implementing changes, 28.65% of surgical interventions commenced between 8:00 and 10:00 a.m., which increased to 32.13% following reorganization, representing an increase of 3.48% (95% CI: 2.1571–4.8013%, χ² = 26.572, *p* < 0.0001). Over 59% of surgical interventions took place between 8:00 a.m. and 12:00 p.m. from January to June 2023, which rose to 64.92% in the first half of 2024, with a difference of 1.55% (95% CI: 0.1415–2.9576%, χ² = 4.652, *p* = 0.0310). The start times of surgical interventions before and after the implementation of measures are detailed in Table 1.

Among the 12 surgical specialties, four were chosen for detailed analysis as they accounted for more than 75% of all surgical interventions, as shown in Figure 2.

Prior to the implementation of measures, the General Surgery department conducted 23.24% of its surgical interventions between 8:00 and 10:00 a.m., and 52.27% between 8:00 a.m. and 12:00 p.m. Following the implementation of measures, these figures rose to 29.26% (difference = 6.02%, 95% CI: 3.6166–8.4217%, χ² = 24.160, *p* < 0.0001) and 56.82% (difference = 3.85%, 95% CI: 1.1303–6.5601%, χ² = 7.695, *p* = 0.0055), respectively, as shown in Figure 3.

In the Plastic Surgery Department, the percentage of surgical interventions conducted between 8:00 and 10:00 a.m. rose from 33.47% before the implementation of measures to 37.86% afterward (difference = 4.39%, 95% CI: 0.5872–8.2019%, χ² = 5.121, *p* = 0.0236). Likewise, the proportion of surgeries performed between 8:00 a.m. and 12:00 p.m. increased from 59.79% to 69.81% (difference = 10.03%, 95% CI: 6.2411–13.7503%, χ² = 26.542, *p* < 0.0001), as detailed in Table 2. The timing of surgical interventions before and after the implementation of these measures is detailed in Table 3.

In the Orthopedics and Traumatology department, the proportion of surgeries conducted between 8:00 and 10:00 a.m. was 23.65% prior to the implementation of measures and increased slightly to 24.97% afterward (difference = 1.32%, 95% CI: −1.1860–3.8117%, χ² = 1.068, *p* = 0.3013). Between 8:00 a.m. and 12:00 p.m., the proportion of surgeries performed changed from 48.70% before to 49.47% after the measures (difference = 0.77%, 95% CI: −2.1436–3.6814%, χ² = 0.268, *p* = 0.6047). The start times of surgical interventions in these intervals, both before and after the implementation of measures across four operating rooms, are shown in Table 4.

In the Urology Department, the percentage of surgeries conducted between 8:00 and 10:00 a.m. rose from 29.56% to 36.39% following the implementation of measures (difference = 6.83%, 95% CI: 3.8617–9.7774%, χ² = 20.278, *p* < 0.0001). Additionally, the proportion of surgeries carried out between 8:00 a.m. and 12:00 p.m. slightly increased from 65.55% to 67.30% (difference = 1.75%, 95% CI: −1.2315–4.7312%, χ² = 1.322, *p* = 0.2503). It should be noted that, both before and after the implementation of measures, these four departments/specialties operated in their own dedicated rooms.

### 3.2. Duration of Surgery 

The reorganization and modernization of the operating theater also involved acquiring state-of-the-art equipment, which enabled more complex interventions and resulted in an increase in the average duration of surgeries from 56.00 min to 65.07 min (difference = 9.07 min, 95% CI: 7.4464–10.6936, t = 10.950, DF = 18,564, *p* < 0.0001), as detailed in Table 5.

Between 8:00 a.m. and 12:00 p.m., the occupancy rate of the operating blocks increased from 63.91% before the implementation of measures to 73.30% after (difference= 9.39%, 95% CI: 1.0848–17.5197%, χ² = 4.902, *p* = 0.0268). The occupancy rate exceeded 75% between 9:00 and 11:00 a.m. both before and after the measures were implemented, rising from 87.53% to 98.07% (difference = 10.54%, 95% CI: 4.0938–17.7662%, χ² = 9.934, *p* = 0.0016), as shown in Figure 4.

Between 8 a.m. and 12 p.m., the general surgery operating room occupancy increased from 63.19% before the implementation of measures to 70.96% after (difference = 7.77%, 95% CI: −0.6362% to 16.0259%, χ² = 3.274, *p* = 0.0704). Before the measures were implemented, occupancy exceeded 75% only between 10 a.m. and 11 a.m., whereas after the measures, it exceeded 75% between 9 a.m. and 12 p.m. The degree of occupancy of the operating blocks before and after the implementation of measures is detailed in Table 6 and Table 7.

The plastic surgery operating room occupancy increased from 65.00% before the implementation of measures to 70.64% afterward (difference = 5.64%, 95% CI: −2.7104% to 13.8829%, χ² = 1.745, *p* = 0.1865). Prior to the measures, occupancy exceeded 75% between 10 a.m. and 12 p.m., whereas after the measures, it exceeded 75% between 9 a.m. and 1 p.m., as shown in Figure 5. Similarly, the orthopedics operating room occupancy rose from 63.19% before the implementation of measures to 74.31% afterward (difference = 7.45%, 95% CI: −0.7038% to 15.4716%, χ² = 3.201, *p* = 0.0736). Before the measures, occupancy exceeded 75% between 10 a.m. and 12 p.m., while after the measures, it did so between 9 a.m. and 1 p.m.

The urology operating room occupancy increased from 55.62% before the implementation of measures to 60.14% afterward (difference = 4.52%, 95% CI: −4.2879% to 13.2297%, χ² = 1.004, *p* = 0.3165). Prior to the measures, occupancy never exceeded 75% during any time interval. However, after the measures were implemented, occupancy surpassed 75% between 9 a.m. and 11 a.m. The occupancy rate of operating theaters from 8:00 a.m. to 12:00 p.m., by operating room, before and after the implementation of measures is detailed in Table 8.

A mixed-effects model was used to account for variability between operating rooms and departments, with surgical duration, room occupancy, and team composition included as random effects. The model confirmed that the new algorithm significantly reduced delays, even when controlling for these factors (*p* < 0.05).

### 3.3. Comparison of Operating Room Utilization Efficiency: Without Algorithm vs. with Computational Algorithm

#### 3.3.1. Occupancy of the Room without a Computational Algorithm (2023)

Distribution of Surgeries: During January–June 2023, the use of operating rooms was relatively uneven. Most operations started between 8:00 and 12:00, with a significant decrease after 12:00. For example, occupancy was 20.42% and 8:00–9:00 and 86.17% between 9:00 and 10:00, but after noon, occupancy fell below 35% for most intervals.

Inefficiencies: Rooms were underutilized after peak hours, and operations were unevenly distributed. This led to periods of inactivity and a suboptimal flow of activity in the operating rooms.

#### 3.3.2. Occupancy of the Room with a Computational Algorithm (2024)

Distribution of Operations: After implementing the measures (assuming the calculation algorithm was applied, according Table 9 and Figure 6), the data for January–June 2024 show a more even distribution of operations throughout the day. Occupancy increased in the 9:00–12:00 time slots, maintaining a more balanced use of the rooms even after peak hours.

Efficiencies:-Optimized Utilization: a significant increase in occupancy was achieved during peak slots, such as from 86.17% in 2023 to 96.31% in 2024 between 9:00 and 10:00, and from 88.88% in 2023 to 99.82% in 2024 between 10:00 and 11:00.-Balanced Distribution: the improved recommendation score led to a better distribution of operations throughout the day, avoiding overcrowding in the morning and underutilization after lunch.-Increased Flexibility: operating rooms were used more efficiently, reducing downtime and allowing greater flexibility in scheduling operations.

The independent samples *t*-test comparing occupancy data from 2023 and 2024 yielded a t-statistic of approximately −0.354 and a *p*-value of about 0.727, indicating no statistically significant difference in operating room occupancy between the two years. This suggests that the variations observed in occupancy between these periods are not substantial enough to be considered statistically meaningful. Before the implementation of an algorithm, the use of operating rooms was uneven, with busy peak periods and underutilized times, leading to inefficiencies and potential delays in scheduling surgeries. We recognize that the hierarchical structure of the data, with surgeries nested within operating rooms or departments, makes a mixed-effects model a suitable approach for analysis. This model would account for random effects and individual differences between departments and operating rooms. While this was not applied in the current analysis, we acknowledge its potential to provide deeper insights. Future studies will explore the use of a mixed-effects model to better control for these nested data structures and improve the robustness of the results.

However, after applying the algorithm, the distribution of operations was optimized, resulting in a more consistent and balanced use of operating rooms throughout the day. This led to a more efficient workflow, reduced downtime, and better utilization of available resources. Overall, the application of a computational algorithm for scheduling and optimizing operating room usage significantly enhanced efficiency, improved task distribution, and streamlined the overall workflow in the operating room.

To provide context for the performance of our proposed scheduling algorithm, we created a comparison table that contrasts it with commonly used scheduling algorithms from the literature. Table 10 highlights the different approaches, key features, advantages, limitations, and potential use cases of each algorithm. This comparison demonstrates how our algorithm, which integrates team coordination and resource management, addresses specific organizational challenges and complements existing methods such as rule-based, optimization, and machine learning approaches. By situating our work within this broader context, we underscore the unique contribution and applicability of our algorithm in improving operating room efficiency.

## 4. Discussion

Improving efficiency and performance remains a constant concern for hospitals, particularly in the face of rising costs [38,39,40]. Ensuring patient safety while maintaining high-quality medical services necessitates continuous monitoring, process streamlining, and the elimination of redundancies [41,42,43]. Operating rooms, as complex environments, require effective management, seamless team coordination, adequate equipment, and thorough patient preparation to function optimally [44].

Driven by cost considerations, medical practice has shifted from sequential to parallel pre-operative stages, such as pre-anesthetic consultations and safety checks [45,46,47,48]. Surgeons play a critical role in standardizing tasks, gathering patient data, and fostering effective team communication, leading to optimal resource utilization and a reduction in avoidable cancellations [49,50]. These efforts align with the quality improvement initiatives of the World Health Organization and other organizations [51,52,53,54].

Our initial analysis focused on factors contributing to extended scheduled surgery times. Despite a staffing schedule from 7 a.m. to 3 p.m., Table 1 indicates that surgeries often extended past 3 p.m. due to emergency cases managed by on-call teams. While the hospital maintains three emergency rooms (general surgery, orthopedics, and urology), these cannot cover all emergencies. Specialties such as plastic surgery, neurosurgery, ophthalmology, and otorhinolaryngology require dedicated operating rooms equipped with specific tools to ensure patient safety. The peak period of activity was identified as 10–11 a.m., with 15.83% of surgeries starting during this time and an 88.88% occupancy rate, despite all rooms being scheduled for surgeries from 8 a.m. Delays in start times, which have been documented in other studies, have prompted proposed solutions [52,55,56,57,58,59,60]. Common issues identified include ineffective communication, rushed preoperative assessments, incomplete patient preparation, missing consent forms, and staff unavailability.

In response, we developed and implemented solutions aimed at minimizing these delays. Hospital management prioritized interdisciplinary teamwork, improved internal communication, individual responsibility, process simplification, data monitoring, and feedback. Dedicated personnel were assigned to oversee schedule management, necessary adjustments, and staff organization. Detailed improvement measures are outlined in the methods section, with data support provided by Medical Statistics and Informatics.

Post-implementation, we observed a statistically significant improvement in surgical start times between 8 a.m. and 10 a.m., increasing from 28.65% to 32.13%. Additionally, there was a non-significant improvement in surgeries scheduled between 8 a.m. and 12 p.m., rising from 59.37% to 64.92%. Similar trends were observed across individual operating rooms.

While surgical duration can impact efficiency [61,62,63], our focus on acquiring new equipment precluded an in-depth analysis of this factor. The average duration stabilized at approximately 60 min, influenced by the complexity of surgeries and patient conditions [64,65,66,67]. Operating room occupancy, calculated as the ratio of used to available surgical time, serves as another metric of efficiency [68,69,70]. We noted an increase in occupancy between 8 a.m. and 12 p.m. (from 63.91% to 73.30%) and between 9 a.m. and 11 a.m. (from 87.53% to 98.07%) following the implementation of measures, although these increases were not statistically significant. All operating rooms achieved over 75% occupancy between 9 a.m. and 11 a.m. after the interventions. Effective scheduling has been shown to contribute significantly to improved occupancy rates [71,72].

The use of computational algorithms in scheduling and optimizing operating room efficiency has shown promising results in other studies as well. For instance, research conducted by Smith et al. demonstrated a significant reduction in downtime and an increase in surgical throughput after implementing an algorithm-based scheduling system, closely mirroring our findings. Similar to our results, their study found that algorithms could effectively balance surgical loads across operating rooms, leading to more consistent room utilization and improved overall efficiency. These parallels reinforce the importance and potential of algorithm-driven solutions in optimizing hospital operations, further validating our approach and suggesting broader applicability across different healthcare settings. The introduction of structured, predictable schedules not only optimized OR usage but also positively influenced team dynamics. Teams reported better communication and reduced conflicts over room availability, contributing to the observed efficiency improvements. The success of this algorithm in our setting suggests that similar computational approaches could be implemented in other resource-limited hospitals to improve efficiency without requiring significant capital investment.

Our study underscores the complexity of enhancing operating room efficiency, which involves procedures, processes, personnel, equipment, infrastructure, and policies. Timely surgical starts are essential and require a multidisciplinary approach. While our study was limited to a single hospital and a few key indicators, the positive outcomes suggest a model that could be broadly implemented. 

To strengthen the scientific value of this study, we plan to compare the new scheduling method with other established methods from the literature, such as block scheduling and priority-based scheduling. To achieve this, we will use discrete event simulation (DES) to model the performance of each method and assess their impact on key metrics, including operating room utilization, on-time surgery starts, and overall efficiency. These methods were selected for their relevance to our hospital’s operational context and their frequent use in previous research. This comparison will provide a more comprehensive evaluation of the effectiveness of the implemented scheduling approach.

This study has several limitations. First, the data were collected from a single hospital, which may limit the generalizability of the findings to other institutions, particularly those with different resource levels or operating room structures. Second, while this study demonstrated improvements in operating room efficiency, the impact on team dynamics was assessed informally and without direct quantitative measures, making it difficult to fully evaluate this aspect. Third, this study focused primarily on operational metrics such as surgical start times and OR utilization, but did not explore patient outcomes, which could provide a more comprehensive understanding of the algorithm’s overall effectiveness. Additionally, the follow-up period of six months may not have been sufficient to fully assess long-term sustainability or the potential fading of the Hawthorne effect, where changes in behavior might occur due to the awareness of being observed. Finally, while the algorithm was validated using historical data, further testing with a more diverse set of surgical cases and settings would strengthen the conclusions drawn.

## 5. Conclusions

The implementation of a computational algorithm, integrated with organizational restructuring, significantly improved operating room efficiency and workflow. By focusing on organizational changes, we created a supportive environment that maximized the algorithm’s effectiveness. This approach not only optimized surgical scheduling but also enhanced team coordination and resource utilization. This study demonstrates that combining operational tools with strategic organizational adjustments can lead to sustainable improvements in healthcare settings. Future research should further explore this integrated strategy, comparing various organizational frameworks rather than focusing solely on algorithmic solutions.

## Figures and Tables

**Figure 1 healthcare-12-01906-f001:**
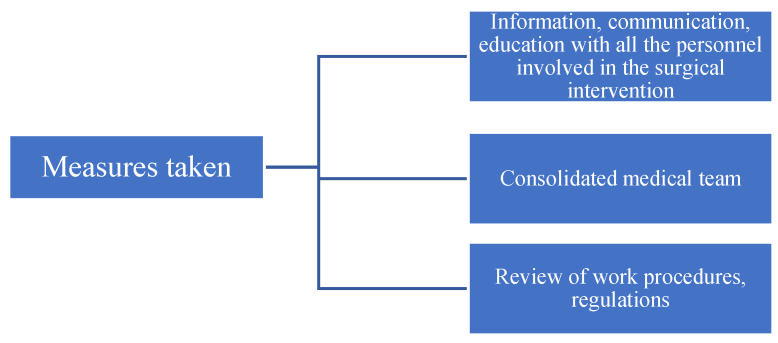
Measures to improve the scheduling of surgical interventions in the operating room.

**Figure 2 healthcare-12-01906-f002:**
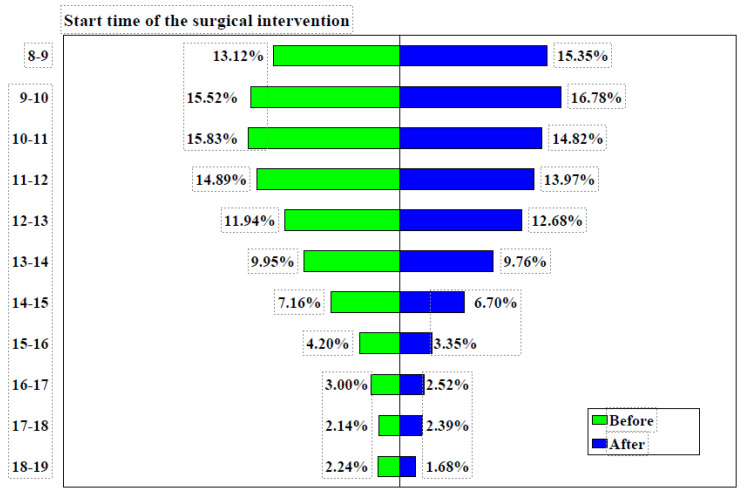
Surgical intervention start times before and after implementation of measures. Note: before = January–June 2023, after = January–June 2024.

**Figure 3 healthcare-12-01906-f003:**
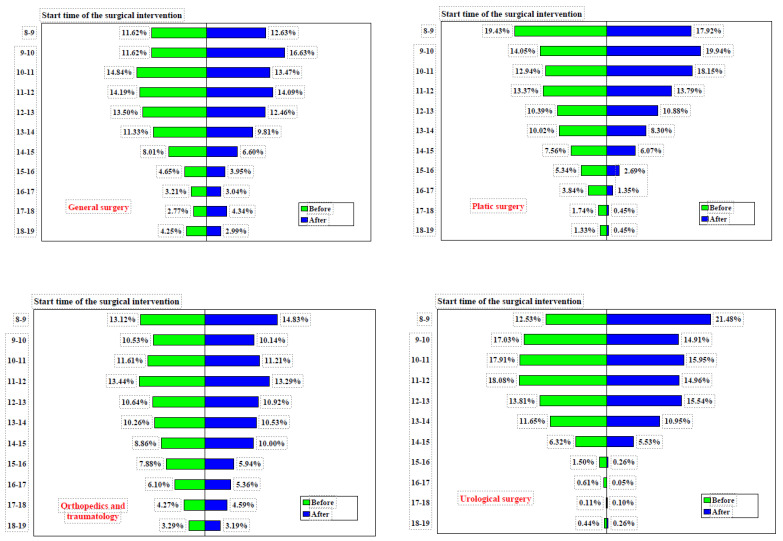
Start time of surgical interventions in four operating rooms before and after implementation of measures. Note: before = Jan–Jun 2023, after = Jan–Jun 2024.

**Figure 4 healthcare-12-01906-f004:**
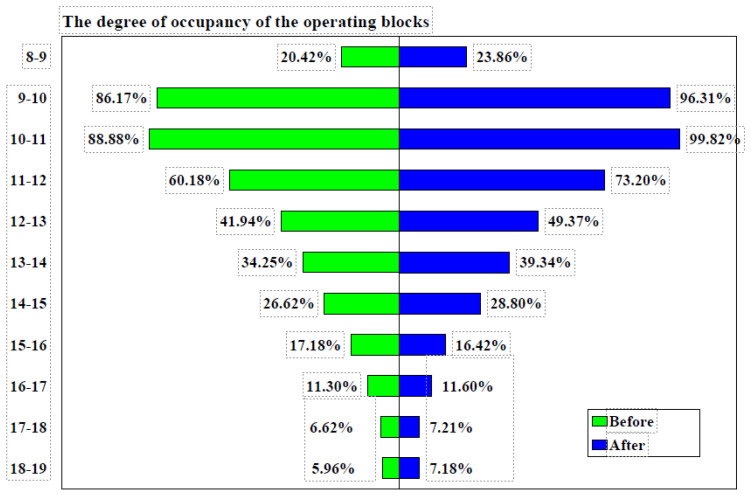
Degree of occupancy of the operating blocks before and after implementation of measures. Note: before = Jan–Jun 2023, after = Jan–Jun 2024.

**Figure 5 healthcare-12-01906-f005:**
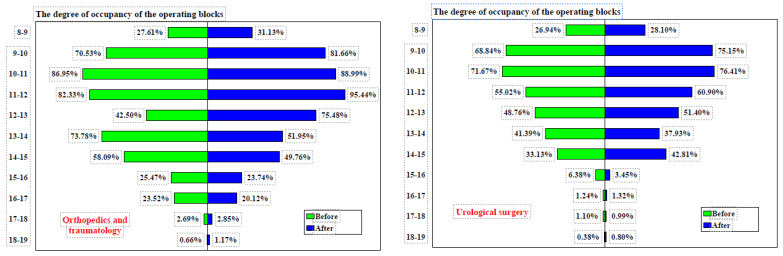
Degree of occupancy of the operating blocks before and after implementation of measures. Note: before = Jan–Jun 2023, after = Jan–Jun 2024.

**Figure 6 healthcare-12-01906-f006:**
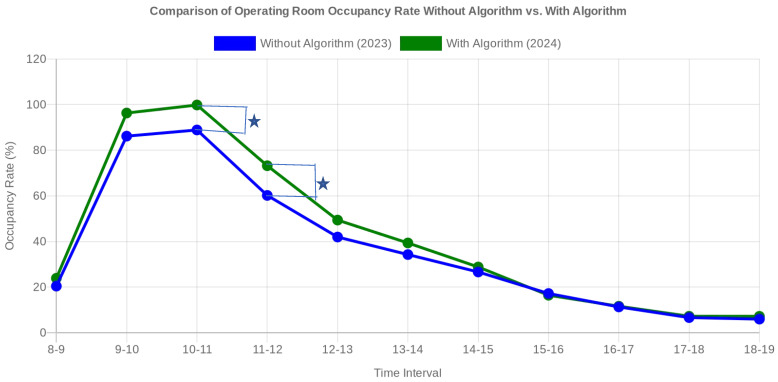
Comparative graph illustrating the degree of occupancy of operating rooms in 2023 (without algorithm) and in 2024 (with algorithm). After the implementation of the algorithm, an increase in the degree of occupancy is observed in most time slots, indicating a more efficient use of the operating rooms. * = statistically significant.

**Table 1 healthcare-12-01906-t001:** Start time of surgical interventions before and after the implementation of measures.

Hours	January–June 2023 (*n* = 4652)	January–June 2024 (*n* = 4631)
8–9	13.12	15.35
9–10	15.52	16.78
10–11	15.83	14.82
11–12	14.89	13.97
12–13	11.94	12.68
13–14	9.95	9.76
14–15	7.16	6.70
15–16	4.20	3.35
16–17	3.00	2.52
17–18	2.14	2.39
18–19	2.24	1.68

*n* = number of surgeries.

**Table 2 healthcare-12-01906-t002:** Start time of surgical interventions in the time intervals 8–10 a.m. and 8–12 p.m., before and after the implementation of measures.

Hours	January–June 2023	January–June 2024	*p*
8–10	28.65	32.13	*p* < 0.0001
8–12	59.37	60.92	*p* = 0.0310

*p* = statistically significance.

**Table 3 healthcare-12-01906-t003:** Surgical intervention start times before and after implementation of measures.

Hours	General Surgery		Plastic Surgery		Orthopedics and Traumatology		Urology	
	Before (*n* = 1352)	After (*n* = 1227)	Before (*n* = 708)	After (*n* = 532)	Before (*n* = 1060)	After (*n* = 1210)	Before (*n* = 935)	After (*n* = 991)
8–9	11.62	12.63	19.43	17.92	13.12	14.83	12.53	21.48
9–10	11.62	16.63	14.05	19.94	10.53	10.14	17.03	14.91
10–11	14.84	13.47	12.94	18.15	11.61	11.21	17.91	15.95
11–12	14.19	14.09	13.37	13.79	13.44	13.29	18.08	14.96
12–13	13.50	12.46	10.39	10.88	10.64	10.92	13.81	15.54
13–14	11.33	9.81	10.02	8.30	10.26	10.53	11.65	10.95
14–15	8.01	6.60	7.56	6.07	8.86	10.00	6.32	5.53
15–16	4.65	3.95	5.34	2.69	7.88	5.94	1.50	0.26
16–17	3.21	3.04	3.84	1.35	6.10	5.36	0.61	0.05
17–18	2.77	4.34	1.74	0.45	4.27	4.59	0.11	0.10
18–19	4.25	2.99	1.33	0.45	3.29	3.19	0.44	0.26

Note: before= Jan–Jun 2023, after= Jan–Jun 2024.

**Table 4 healthcare-12-01906-t004:** Start time of surgical interventions in the time intervals 8–10 a.m. and 8–12 p.m., before and after the implementation of measures, in four operating rooms.

Specialty	Hours	January–June 2023	January–June 2024	*p*
General surgery	8–10	23.24	29.26	*p* < 0.0001
	8–12	52.27	56.82	*p* = 0.0055
Plastic surgery	8–10	33.47	37.86	*p* = 0.0236
	8–12	59.78	69.81	*p* < 0.0001
Orthopedics and traumatology	8–10	23.65	24.97	*p* = 0.3013
	8–12	48.70	49.47	*p* = 0.6047
Urology	8–10	29.56	36.39	*p* < 0.0001
	8–12	65.55	67.30	*p* = 0.2503

*p* = statistically significance.

**Table 5 healthcare-12-01906-t005:** Duration of surgical interventions before and after implementation of measures.

	January–June 2023 (*n* = 4652)	January–June 2024 (*n* = 4361)	*p*
Duration (minutes)	56.00 ± 52.64	65.07 ± 60.00	*p* < 0.0001

*p* = statistically significance.

**Table 6 healthcare-12-01906-t006:** Degree of occupancy of the operating blocks before and after implementation of measures according to the hours.

Hours	January–June 2023	January–June 2024	*p*	*p*
8–9	20.42	23.86	*p* = 0.0268	
9–10	86.17	96.31		*p* = 0.0016
10–11	88.88	99.82		
11–12	60.18	73.20		
12–13	41.94	49.37		
13–14	34.25	39.34		
14–15	26.62	28.80		
15–16	17.18	16.42		
16–17	11.30	11.60		
17–18	6.62	7.21		
18–19	5.96	7.18		

*p* = statistically significance.

**Table 7 healthcare-12-01906-t007:** Degree of occupancy of the operating blocks before and after implementation of measures.

Hours	General Surgery		Plastic Surgery		Orthopedics and Traumatology		Urology	
	Before	After	Before	After	Before	After	Before	After
8–9	20.39	25.22	18.66	21.98	27.61	31.13	26.94	28.10
9–10	73.51	82.86	72.15	83.84	70.53	81.66	68.84	75.15
10–11	80.66	88.26	87.10	92.15	86.95	88.99	71.67	76.41
11–12	78.21	87.49	82.10	84.60	82.33	95.44	55.02	60.90
12–13	42.22	71.22	68.31	75.50	42.50	75.48	48.76	51.40
13–14	35.58	22.77	42.30	41.90	73.78	51.95	41.39	37.93
14–15	19.15	30.70	8.84	7.79	58.09	49.76	33.13	42.81
15–16	13.17	21.78	3.26	3.82	25.47	23.74	6.38	3.45
16–17	8.94	14.79	2.13	0.87	23.52	20.12	1.24	1.32
17–18	5.46	8.46	0.37	0.54	2.69	2.85	1.10	0.99
18–19	1.24	4.27	0.19	0.30	0.66	1.17	0.38	0.80

Note: before = Jan–Jun 2023, after = Jan–Jun 2024.

**Table 8 healthcare-12-01906-t008:** Occupancy rate of operating theaters from 8:00 a.m. to 12:00 p.m., by operating room, before and after implementation of measures.

Specialty	Hours	January–June 2023	January–June 2024	*p*
General surgery	8–12	63.19	70.96	0.0704
Plastic surgery	8–12	65.00	70.64	0.1865
Orthopedics and traumatology	8–12	66.86	74.31	0.0736
Urology	8–12	55.62	60.14	0.3165

*p* = statistically significance.

**Table 9 healthcare-12-01906-t009:** Comparison of operating room occupancy without and with algorithm for 2023 and 2024.

Time Interval	Occupancy Rate 2023 (%)	Occupancy Rate 2024 (%)	*p*	t	*p*
8–9	20.42	23.86	0.125	−0.354	0.727
9–10	86.17	96.31	0.050		
10–11	88.88	99.82	0.037 *		
11–12	60.18	73.20	0.021 *		
12–13	41.94	49.37	0.067		
13–14	34.25	39.34	0.147		
14–15	26.62	28.80	0.315		
15–16	17.18	16.42	0.419		
16–17	11.30	11.60	0.674		
17–18	6.62	7.21	0.289		
18–19	5.96	7.18	0.051		

*p* = statistically significance, t = *t* Student test, * = correlation is significant at the 0.05 level (2-tailed).

**Table 10 healthcare-12-01906-t010:** Comparison of the proposed scheduling algorithm with existing operating room scheduling algorithms.

Algorithm	Approach	Key Features	Advantages	Limitations	Potential Use Cases	References
Proposed Algorithm	Custom Scheduling Algorithm	Incorporates team coordination and resource utilization	Enhances workflow efficiency and team dynamics	May require customization for different settings	Hospitals with specific team and resource management needs	[Our study]
First Come, First Served (FCFS)	Rule-based	Simple, processes cases as they arrive	Easy to implement, no complex calculations	Does not account for priority or resource constraints	Small hospitals with low case complexity	[29]
Block Scheduling	Rule-based	Reserves specific time slots for departments	Predictable schedule, easy resource allocation	Can lead to underutilization during blocks	Hospitals with regular, predictable caseloads	[30]
Mixed-Integer Linear Programming (MILP)	Optimization	Solves for optimal allocation of resources and time	Finds globally optimal solutions	Computationally intensive, complex to implement	Large hospitals with diverse and complex caseloads	[31]
Genetic Algorithms	Heuristic/Metaheuristic	Evolves solutions over iterations	Can handle large, complex problem spaces	May not always find the optimal solution, sensitive to parameter tuning	Complex, variable scheduling environments	[32]
Simulated Annealing	Heuristic/Metaheuristic	Searches solution space with probabilistic decisions	Can escape local optima, flexible	May require many iterations, sensitive to parameters	Dynamic environments with changing constraints	[33]
Priority-Based Scheduling	Rule-based	Prioritizes cases based on urgency or other criteria	Efficient for handling emergency cases	Can lead to delays for non-priority cases	Hospitals with high volume of emergency cases	[34]
Stochastic Optimization	Probabilistic/Optimization	Considers variability in surgery durations	Accounts for uncertainty and variability	Requires accurate probability distributions	Environments with high variability in case lengths	[35]
Neural Networks	Machine Learning	Learns patterns from historical data	Adaptive to changes, can predict outcomes	Requires large datasets, less interpretable	Hospitals with extensive historical data	[36]
Reinforcement Learning	Machine Learning	Learns optimal scheduling policies through trial and error	Can adapt to dynamic and complex environments	Needs extensive training, may require simulation	Highly dynamic environments with changing demands	[37]

## Data Availability

All the data processed in this article are part of the research for a doctoral thesis, being archived in the aesthetic medical office, where the interventions were performed.

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
