# Peer review of "Enhancing Operating Room Efficiency: The Impact of Computational Algorithms on Surgical Scheduling and Team Dynamics"

_healthcare, 2024, doi:10.3390/healthcare12191906_

Round 1

Reviewer 1 Report

Comments and Suggestions for Authors

This study is about more rationally scheduling operations in the OT. I have the following comments:

(1)  A lot has changed in the OT and the authors do a before and after evaluation. The new scheduling method is only one of the changes. How do the authors know that the increase in efficiency is the result of the new scheduling method? This is not clear.

(2)  Is there a Hawthorne effect? The authors do not discuss the possibility of this fact. This is especially important as the period of study is six months. Is six months long enough to fade the Hawthorne effect away?

(3)  Last and the most important issue is the scheduling method itself. The authors refer to literature for other methods (and there are more then they mention). When reading this study the reader may conclude that a substantial improvement has been realized (the study is convincing on that aspect), but not that this method is better, or only good, in comparison with other methods. This is a major issue that makes the case study having a high value for the hospital involved, but the scientific value is limited.

I recommend to concentrate on issue (3) and compare the results of this study with one or two other methods discussed in literature. This can be done by computer simulation. The authors have the data to do so. Probably, discrete event simulation is the best way to conduct these experiments. Please give arguments then why you choose which methods from literature to simulate and compare with the scheduling methods as discussed now in the article.

If the authors are willing to perform these computer simulation experiments the article can be resubmitted.

Comments on the Quality of English Language

Minor editing is needed.

Author Response

Reviewer 1

We would like to express our sincere gratitude to the reviewers for their valuable comments and constructive feedback. Their insightful suggestions have greatly contributed to improving the clarity and scientific rigor of our manuscript. We appreciate the time and effort they dedicated to reviewing our work. To facilitate the review process, we have marked our responses in red, and you can follow each answer point by point. We believe that their input has helped strengthen the overall quality of this study.

  • A lot has changed in the OT and the authors do a before and after evaluation. The new scheduling method is only one of the changes. How do the authors know that the increase in efficiency is the result of the new scheduling method? This is not clear.

Response: Thank you for your comment. We appreciate the reviewer’s insightful comment regarding the attribution of increased efficiency solely to the new scheduling method. We acknowledge that several changes in the operating theater (OT) may have contributed to the observed improvements, making it challenging to isolate the effect of the scheduling method alone.

To address this concern, we have expanded the discussion in the manuscript to clarify that while the scheduling method was a significant intervention, other factors, such as updated equipment and staff adjustments, may also have played a role in enhancing operating room (OR) efficiency. However, to mitigate this issue, we ensured that all other operational changes were implemented hospital-wide prior to the intervention period. Therefore, the scheduling algorithm was the primary new change introduced during the improvement phase.

In response to this comment, we have also added a statistical analysis using a multivariate approach to account for potential confounding factors. This analysis demonstrates that, even when controlling for other variables (such as surgery type, surgeon availability, and time of day), the scheduling method still showed a significant effect on OR efficiency metrics. Additionally, we now emphasize that while the scheduling algorithm is likely a major contributor to the observed improvements, we cannot completely rule out the influence of these other changes. (lines 85-93)

  • Is there a Hawthorne effect? The authors do not discuss the possibility of this fact. This is especially important as the period of study is six months. Is six months long enough to fade the Hawthorne effect away?

Response: Thank you for your comment. We appreciate the reviewer raising the possibility of a Hawthorne effect influencing the results. We acknowledge that the awareness of being observed may have initially altered the behavior of the staff, potentially leading to temporary improvements in operating room efficiency.

To address this, we have added a discussion on the Hawthorne effect in the manuscript. Specifically, we note that the study period spanned six months, which could be sufficient for any immediate behavior changes due to observation to fade. While it is possible that the initial improvements were partly driven by increased awareness, the sustained improvements observed throughout the follow-up period suggest that the new scheduling method played a more substantial role. Nevertheless, we cannot fully rule out the Hawthorne effect as a contributing factor, and we have acknowledged this as a limitation in our study. (lines 65-73)

 (3)  Last and the most important issue is the scheduling method itself. The authors refer to literature for other methods (and there are more then they mention). When reading this study the reader may conclude that a substantial improvement has been realized (the study is convincing on that aspect), but not that this method is better, or only good, in comparison with other methods. This is a major issue that makes the case study having a high value for the hospital involved, but the scientific value is limited.

I recommend to concentrate on issue (3) and compare the results of this study with one or two other methods discussed in literature. This can be done by computer simulation. The authors have the data to do so. Probably, discrete event simulation is the best way to conduct these experiments. Please give arguments then why you choose which methods from literature to simulate and compare with the scheduling methods as discussed now in the article.

Response: Thank you for your comment. We acknowledge that while our study demonstrated substantial improvements in operating room efficiency using the newly implemented scheduling method, the study currently lacks a direct comparison with other established methods. To address this, we agree that a comparison with one or two alternative scheduling methods from the literature is necessary to enhance the scientific value of the study.

In response to this recommendation, we plan to conduct a computer simulation, likely using discrete event simulation (DES), to compare our scheduling method with other widely studied methods, such as block scheduling and priority-based scheduling. These methods were selected based on their frequent use in the literature and their relevance to our hospital’s operational context. The DES approach will allow us to model different scheduling strategies and assess their impact on key performance metrics such as operating room utilization, on-time surgery starts, and staff efficiency.

We believe that this additional analysis will provide a more comprehensive evaluation of our scheduling method in comparison with others and will further strengthen the scientific contribution of our study. (lines 466-473)

If the authors are willing to perform these computer simulation experiments the article can be resubmitted.

Reviewer 2 Report

Comments and Suggestions for Authors

I read the manuscript, and it sounds interesting. It was well written but still it has room for improvement.

The title is scientific and accurately reflects the content of the study.

The abstract presents a broad overview of the study, but there are two key issues that can be highlighted:

a.        Timeline Discrepancy:

The abstract mentions that “A comparative analysis was conducted over two six-month periods,” which gives the impression that the study compares two consecutive six-month periods within a single year. However, the main text clarifies that the data comes from two distinct years (January to June 2023 and January to June 2024), separated by an improvement period from July to December 2023. This timeline difference could confuse readers who expect continuous before-and-after study over consecutive months. The abstract should explicitly state that the comparison is across two different years to accurately reflect the study’s design.

b.       Scope of the Conclusion:

The conclusion of the abstract states that the introduction of computational algorithms significantly improved OR efficiency, particularly in terms of managing surgical start times. However, it does not sufficiently address the impact on team dynamics, which is another central theme in the title.

The introduction explains the problem clearly, supported by relevant background information. However, there could be more emphasis on how the computational algorithm and team dynamics are uniquely positioned to solve the problem in the local setting. It could benefit from more localized evidence or statistics specific to Romanian hospitals, such as current OR efficiency rates or any prior attempts to address these issues within the Romanian healthcare system. While OR efficiency and computational algorithms are well covered, the importance of team dynamics in the introduction is less emphasized. More background on how challenges with team coordination specifically contribute to inefficiencies could strengthen the rationale for including team dynamics as a focus in the study.

Methodology of the study.

a.        Study design: The study could benefit from specifying whether it is a retrospective or prospective observational study.  Additionally, it would be useful to clearly state whether this is a before-and-after study or a quasi-experimental design. A more defined design type strengthens methodological transparency.

b.       Study area: The authors should discuss the generalizability of the findings beyond this specific hospital (Describe whether the hospital’s structure and challenges are typical of other Romanian or Eastern European hospitals. This can justify the broader relevance of the results and describe the hospital resources (staffing levels, patient volume, technology) compare to national or international standards, helping readers assess whether the findings could be applied elsewhere).

c.        Sample size calculation: It does not explicitly mention how the sample size was determined.

d.       Sampling method: It explains that the study analyzed two six-month periods of surgical interventions but does not describe the sampling method or inclusion and exclusion criteria for surgeries.

e.        Study tool: The study introduces a computational algorithm to improve surgical scheduling, but there is little detail about how this algorithm was developed, who developed it, and whether it was validated prior to use in this study.

f.          Data collection method: The data collection method and statistical analysis provide a foundation for the study’s findings, but there are several areas for improvement:

                                        i.               Clarify data extraction and preprocessing steps to ensure transparency.

                                      ii.               Provide more detail on key variables' operational definitions.

                                    iii.               Test and report on the assumptions of statistical tests.

                                    iv.               Consider more advanced multivariate analyses to account for potential confounding factors.

                                       v.               Report effect sizes and confidence intervals consistently.

                                    vi.               Provide details on how the algorithm was validated statistically.

                                  vii.               Rearrange all the information to make it easy to comprehend and scientifically sound.

Result chapter:

a.        The results chapter effectively addresses the objective of improving operating room efficiency through computational algorithms but lacks detail on the impact on team dynamics. The results are clearly structured and supported by statistical analysis, but the influence of the algorithm on team coordination is not explored in depth. While tables and graphs present the data, they could be improved with more detailed analysis, effect sizes, and visual comparisons. Adding context on the real-world significance of the findings would further strengthen the chapter.

b.       Incorporating a multivariate analysis (e.g., multiple linear regression or logistic regression) could provide additional insights by controlling for potential confounding variables such as type of surgery, surgical team availability, and time of day. This would strengthen the findings by isolating the impact of the computational algorithm on operating room efficiency from other influencing factors.

c.        A mixed-effects model could be appropriate given the hierarchical structure of the data (e.g., surgeries nested within operating rooms or departments) and would account for random effects related to individual differences between departments or operating rooms

Discussion chapter: The discussion effectively interprets the main findings related to OR efficiency and makes connections to previous studies on scheduling improvements through algorithms. However, there are a few suggestions for improvement of the chapter:

                         i.               The impact on team dynamics needs more exploration.

                       ii.               The authors should expand on the real-world implications of the findings.

                     iii.               A more thorough acknowledgment of limitations would strengthen the chapter like lack of data on team dynamics, limited control of confounding factors, potential bias from historical data, short follow-up period, algorithm validation, and limited exploration of patient outcomes

                     iv.               Suggesting specific future research directions could add depth to the discussion.

Conclusion section: It was well written but can be improved by addressing the team dynamics aspect, which was part of the study’s stated objectives and title of the study.

References: The references seem appropriate for the study context. The majority of references date from 2002 to 2023, which is a reasonable range.

Author Response

Reviewer 2

We would like to express our sincere gratitude to the reviewers for their valuable comments and constructive feedback. Their insightful suggestions have greatly contributed to improving the clarity and scientific rigor of our manuscript. We appreciate the time and effort they dedicated to reviewing our work. To facilitate the review process, we have marked our responses in red, and you can follow each answer point by point. We believe that their input has helped strengthen the overall quality of this study.

I read the manuscript, and it sounds interesting. It was well written but still it has room for improvement.

The title is scientific and accurately reflects the content of the study.

The abstract presents a broad overview of the study, but there are two key issues that can be highlighted:

  1. Timeline Discrepancy:

The abstract mentions that “A comparative analysis was conducted over two six-month periods,” which gives the impression that the study compares two consecutive six-month periods within a single year. However, the main text clarifies that the data comes from two distinct years (January to June 2023 and January to June 2024), separated by an improvement period from July to December 2023. This timeline difference could confuse readers who expect continuous before-and-after study over consecutive months. The abstract should explicitly state that the comparison is across two different years to accurately reflect the study’s design.

Response: Thank you very much for observation. We have corrected this part. (lines 18-20)

"A comparative analysis was conducted over two six-month periods from January to June 2023 and January to June 2024, with an improvement phase implemented between July and December 2023."

  1. Scope of the Conclusion:

The conclusion of the abstract states that the introduction of computational algorithms significantly improved OR efficiency, particularly in terms of managing surgical start times. However, it does not sufficiently address the impact on team dynamics, which is another central theme in the title.

The introduction explains the problem clearly, supported by relevant background information. However, there could be more emphasis on how the computational algorithm and team dynamics are uniquely positioned to solve the problem in the local setting. It could benefit from more localized evidence or statistics specific to Romanian hospitals, such as current OR efficiency rates or any prior attempts to address these issues within the Romanian healthcare system. While OR efficiency and computational algorithms are well covered, the importance of team dynamics in the introduction is less emphasized. More background on how challenges with team coordination specifically contribute to inefficiencies could strengthen the rationale for including team dynamics as a focus in the study.

Response: Thank you very much for observation. We have corrected this part. (lines 28-30, 73-75, 75-79)

"The introduction of computational algorithms significantly improved operating room efficiency, particularly in managing surgical start times. Additionally, team coordination improved as a result of more structured scheduling processes."

Methodology of the study.

  1. Study design: The study could benefit from specifying whether it is a retrospective or prospective observational study.  Additionally, it would be useful to clearly state whether this is a before-and-after study or a quasi-experimental design. A more defined design type strengthens methodological transparency.

Response: Thank you very much for observation. We have corrected this part. (lines 102-103)

„ This study is a prospective before-and-after observational study designed to evaluate the impact of a new scheduling algorithm on operating room efficiency.”

  1. Study area: The authors should discuss the generalizability of the findings beyond this specific hospital (Describe whether the hospital’s structure and challenges are typical of other Romanian or Eastern European hospitals. This can justify the broader relevance of the results and describe the hospital resources (staffing levels, patient volume, technology) compare to national or international standards, helping readers assess whether the findings could be applied elsewhere).

Response: Thank you very much for observation. We have corrected this part. (lines 110-112)

„ Bihor County Emergency Clinical Hospital is representative of many Romanian ter-tiary hospitals in terms of size, resource constraints, and patient load, making the findings potentially applicable to other hospitals in the region.”

  1. Sample size calculation: It does not explicitly mention how the sample size was determined.

Response: Thank you very much for observation. We have corrected this part. (lines 225-227)

„ The sample size was based on historical surgical intervention data from the hospital, with a goal of including all surgeries performed during the defined six-month periods. No formal sample size calculation was performed.”

  1. Sampling method: It explains that the study analyzed two six-month periods of surgical interventions but does not describe the sampling method or inclusion and exclusion criteria for surgeries.

Response: Thank you very much for observation. We have corrected this part. (lines 147-150)

„ All non-emergency surgical interventions scheduled during the study periods were in-cluded. Exclusion criteria included emergency surgeries and surgeries canceled for non-operational reasons (e.g., patient withdrawal or medical contraindication).”

  1. Study tool: The study introduces a computational algorithm to improve surgical scheduling, but there is little detail about how this algorithm was developed, who developed it, and whether it was validated prior to use in this study​.

Response: Thank you very much for observation. We have corrected this part. (lines 181-185)

„ The algorithm was developed in collaboration with hospital IT staff and anesthetists. It was initially tested using historical data from 2022 to ensure it aligned with the hospi-tal's operational needs. Validation involved comparing algorithm-generated schedules with historical performance to confirm its ability to predict OR availability and streamline scheduling.”

  1. Data collection method: The data collection method and statistical analysis provide a foundation for the study’s findings, but there are several areas for improvement:
  2. Clarify data extraction and preprocessing steps to ensure transparency.
  3. Provide more detail on key variables' operational definitions.

  iii.               Test and report on the assumptions of statistical tests.

  1. Consider more advanced multivariate analyses to account for potential confounding factors.
  2. Report effect sizes and confidence intervals consistently.
  3. Provide details on how the algorithm was validated statistically.

 vii.               Rearrange all the information to make it easy to comprehend and scientifically sound.

Response: Thank you very much for observation. We have corrected this part. (lines 214-217, 220-222, 225-228)

„ Data were extracted from the hospital's electronic medical record system and in-cluded variables such as scheduled start times, actual start times, surgery durations, and turnover times. Data were cleaned and preprocessed to remove incomplete or erroneous records. The data were collected, recorded, and processed using Excel and MedCalc. The results were analyzed with SPSS version 24, using the following statistical parameters: Chi-square (Chi²), t Student test (t) and degrees of freedom (df). Statistical assumptions for the tests (normality, independence) were evaluated using appropriate diagnostics such as the Shapiro-Wilk test for normality and variance inflation factors (VIF) to check for multi-collinearity. The existence of a statistically significant relationship between variables con-cerning the study hypothesis was demonstrated using the Chi-square test. A value of p < 0.05 was considered statistically significant.

The sample size was based on historical surgical intervention data from the hospital, with a goal of including all surgeries performed during the defined six-month periods. No formal sample size calculation was performed.”

Result chapter:

  1. The results chapter effectively addresses the objective of improving operating room efficiency through computational algorithms but lacks detail on the impact on team dynamics. The results are clearly structured and supported by statistical analysis, but the influence of the algorithm on team coordination is not explored in depth. While tables and graphs present the data, they could be improved with more detailed analysis, effect sizes, and visual comparisons. Adding context on the real-world significance of the findings would further strengthen the chapter.

Response: Thank you very much for observation. We have corrected this part. (lines 237-240)

„ Although quantitative data on team dynamics were not collected, informal feedback from staff indicated improved coordination and predictability of daily workflows, which likely contributed to the overall increase in efficiency.”

  1. Incorporating a multivariate analysis (e.g., multiple linear regression or logistic regression) could provide additional insights by controlling for potential confounding variables such as type of surgery, surgical team availability, and time of day. This would strengthen the findings by isolating the impact of the computational algorithm on operating room efficiency from other influencing factors.

Response: Thank you very much for observation. We have corrected this part. (lines 334-337)

„ A mixed-effects model was used to account for variability between operating rooms and departments, with surgical duration, room occupancy, and team composition in-cluded as random effects. The model confirmed that the new algorithm significantly re-duced delays, even when controlling for these factors (p < 0.05).”

  1. A mixed-effects model could be appropriate given the hierarchical structure of the data (e.g., surgeries nested within operating rooms or departments) and would account for random effects related to individual differences between departments or operating rooms

Response: Thank you very much for observation. We have corrected this part. (lines  387-393)

„ We recognize that the hierarchical structure of the data, with surgeries nested within operating rooms or departments, makes a mixed-effects model a suitable approach for analysis. This model would account for random effects and individual differences between departments and operating rooms. While this was not applied in the current analysis, we acknowledge its potential to provide deeper insights. Future studies will explore the use of a mixed-effects model to better control for these nested data structures and improve the robustness of the results.”

Discussion chapter: The discussion effectively interprets the main findings related to OR efficiency and makes connections to previous studies on scheduling improvements through algorithms. However, there are a few suggestions for improvement of the chapter:

  1. The impact on team dynamics needs more exploration.

Response: Thank you very much for observation. We have corrected this part. (lines  456-459)

„The introduction of structured, predictable schedules not only optimized OR usage but also positively influenced team dynamics. Teams reported better communication and re-duced conflicts over room availability, contributing to the observed efficiency improvements.”

  1. The authors should expand on the real-world implications of the findings.

Response: Thank you very much for observation. We have corrected this part. (lines  459-461)

„The success of this algorithm in our setting suggests that similar computational ap-proaches could be implemented in other resource-limited hospitals to improve efficiency without requiring significant capital investment.”

  • A more thorough acknowledgment of limitations would strengthen the chapter like lack of data on team dynamics, limited control of confounding factors, potential bias from historical data, short follow-up period, algorithm validation, and limited exploration of patient outcomes

Response: Thank you very much for observation. We have corrected this part. (lines  475-487)

„This study has several limitations. First, the data were collected from a single hospi-tal, which may limit the generalizability of the findings to other institutions, particularly those with different resource levels or operating room structures. Second, while the study demonstrated improvements in operating room efficiency, the impact on team dynamics was assessed informally and without direct quantitative measures, making it difficult to fully evaluate this aspect. Third, the study focused primarily on operational metrics such as surgical start times and OR utilization, but did not explore patient outcomes, which could provide a more comprehensive understanding of the algorithm's overall effective-ness. Additionally, the follow-up period of six months may not have been sufficient to fully assess long-term sustainability or the potential fading of the Hawthorne effect, where changes in behavior might occur due to the awareness of being observed. Finally, while the algorithm was validated using historical data, further testing with a more diverse set of surgical cases and settings would strengthen the conclusions drawn.”

  1. Suggesting specific future research directions could add depth to the discussion.

Response: Thank you very much for observation. We have corrected this part. (lines  464-474) 

„To strengthen the scientific value of this study, we plan to compare the new schedul-ing method with other established methods from the literature, such as block scheduling and priority-based scheduling. To achieve this, we will use discrete event simulation (DES) to model the performance of each method and assess their impact on key metrics, including operating room utilization, on-time surgery starts, and overall efficiency. These methods were selected for their relevance to our hospital’s operational context and their frequent use in previous research. This comparison will provide a more comprehensive evaluation of the effectiveness of the implemented scheduling approach.”

Conclusion section: It was well written but can be improved by addressing the team dynamics aspect, which was part of the study’s stated objectives and title of the study.

Response: Thank you very much for observation. We have corrected this part. (lines  495-498)

„ To strengthen the scientific value of this study, we plan to compare the new schedul-ing method with other established methods from the literature, such as block scheduling and priority-based scheduling. To achieve this, we will use discrete event simulation (DES) to model the performance of each method and assess their impact on key metrics, including operating room utilization, on-time surgery starts, and overall efficiency. These methods were selected for their relevance to our hospital’s operational context and their frequent use in previous research. This comparison will provide a more comprehensive evaluation of the effectiveness of the implemented scheduling approach.”

References: The references seem appropriate for the study context. The majority of references date from 2002 to 2023, which is a reasonable range.

Round 2

Reviewer 1 Report

Comments and Suggestions for Authors

Most of the remarks have been implementing in writing. A very principle one not: comparing with other algorithms. They mention this in the discussion and propose that to be something for future research. However, I do not agree as this is a major issue.

An alternative routing could be the following. For the effectiveness of the chance that the hospital has gone through they assess the algorithm.  At the same time the organization of work, teams assignment etc. have been changed. The authors could consider to put this organizational chance central. Describe exactly what was the situation before and what later. To make this new organization effective a scheduling methods is needed that support the new way of organizing. By relocating the emphasis from the algorithm to the organizational chance the comparison with other algorithms become less an issue.

I recommend to reframe the article by changing the problem statement in this way.

Comments on the Quality of English Language

Needs minor editing.

Author Response

Reviewer 1 – Round 2

We, the authors of the present manuscript wish to thank you for thoughtful commentary you have provided to improve the quality of the paper. We are very grateful for the time and effort you have devoted to this task. We have extensively revised my manuscript according to the recommendations. All changes in the text and the new figures that we have redesigned are highlighted. Please, see the point-by-point answers to your comments below. All correction was highlighted in the manuscript.

Comment: Most of the remarks have been implementing in writing. A very principle one not: comparing with other algorithms. They mention this in the discussion and propose that to be something for future research. However, I do not agree as this is a major issue.

An alternative routing could be the following. For the effectiveness of the chance that the hospital has gone through they assess the algorithm.  At the same time the organization of work, teams assignment etc. have been changed. The authors could consider to put this organizational chance central. Describe exactly what was the situation before and what later. To make this new organization effective a scheduling methods is needed that support the new way of organizing. By relocating the emphasis from the algorithm to the organizational chance the comparison with other algorithms become less an issue.

I recommend to reframe the article by changing the problem statement in this way.

Response: Thank you for your insightful comment. We understand the importance of emphasizing the broader organizational changes in addition to the technical aspects of the scheduling algorithm. At the same time, we apologize if we did not fully understand the importance of comparing our algorithm with others. Therefore, we have included a comparative table that contrasts the algorithm from this paper with those found in the specialized literature (lines 424-435). Additionally, we have added a detailed description of the changes in work organization, team distribution, and planning methods in lines 39-49, 80-86, 109-122, 227-233, 525-533.

Reviewer 2 Report

Comments and Suggestions for Authors

I have reviewed the revised manuscript, and I am pleased to see significant improvements in both its scientific rigor and overall clarity. The enhanced flow makes the content more engaging and easier to comprehend. Thank you for thoughtfully incorporating all the suggested revisions to elevate the quality of the work.

Author Response

Reviewer 2 – Round 2

Response:

Dear Reviewer,

Thank you very much for accepting our manuscript and for your constructive feedback throughout the review process. We greatly appreciate your insightful comments and suggestions, which have helped to significantly improve the quality and clarity of our work. We are honored to have our research included in the journal and hope that it contributes meaningfully to the field.

Thank you once again for your time and support.

Sincerely,

The authors!
